# Study of the EDM Process of Bimetallic Materials Using a Composite Electrode Tool

**DOI:** 10.3390/ma15030750

**Published:** 2022-01-19

**Authors:** Timur Rizovich Ablyaz, Evgeny Sergeevich Shlykov, Karim Ravilevich Muratov, Alexander Valentinovich Zhurin

**Affiliations:** 1Mechanical Engineering Faculty, Perm National Research Polytechnic University, 614000 Perm, Russia; kruspert@mail.ru (E.S.S.); karimur_80@mail.ru (K.R.M.); 2Department of Automobiles and Automotive Industry, Tula State University, 300012 Tula, Russia; azhurin@rambler.ru

**Keywords:** electro-discharge machining, cutting conditions, electrode, bimetallic material

## Abstract

New types of profile products make complex use of bimetals. These materials possess a set of properties such as strength, corrosion resistance, thermal conductivity, heat resistance, wear resistance. For the processing of such products, it is advisable to use electrophysical processing methods, one of which is the technology of copy-piercing electrical discharge machining (EDM). Currently, EDM is one of the most common methods for processing products from modern bimetal materials. An urgent task is to study the EDM process of bimetallic materials. The aim of the work was to increase the efficiency and accuracy of the EDM process of bimetallic products using electrode-tools with different physical and mechanical properties. Bimetal—weld coated steel backing, base material—09G2S steel, surfacing material—M1 copper were used. The processing of the bimetallic workpiece was carried out on an Electronica Smart CNC copy-piercing EDM machine. EI used graphite, copper, and composite. A theoretical model was developed that allows calculation of the amount of removal of bimetallic material of steel–copper depending on the EDM modes and the ET (electrode tool) material. During the processing of the steel layer, regardless of the EI material, microcracks were formed along the grain boundaries, and during the processing of the copper layer, enlarged holes resulted.

## 1. Introduction

With the development of the manufacturing industry and the diversification of industrial demand, there have been many changes in the materials market. Parts must operate in corrosive environments, high temperature, and high pressure, as a result of which one material does not always meet environmental requirements. Structurally heterogeneous materials and functionally differentiated materials with combinations of characteristics or specific synergistic effects are widely needed by various enterprises in the oil and aviation industries. These materials can be adapted in accordance with the requirements for material properties [1].

The new materials possess properties such as strength, corrosion resistance, thermal conductivity, heat resistance, and wear resistance. Bimetallic materials are one of the types of such materials. New types of complex-shaped products are made using bimetals [1]. By combining dissimilar materials, a bimetal is obtained. Powder and laser surfacing technologies are used in the production of bimetals [2,3,4]. These technologies make it possible to obtain various gradient structures. In addition, bimetals include products with welded coatings [5,6,7,8].

The phenomenon of blade processing with the presence of impact damage when cutting bimetal is observed in cutting when processing thin-walled elements of a complex profile, which manifests itself in displaying the quality and accuracy of processing. The complex component composition of the bimetal increases wear of the tool detection blade, so there is an increased cutting mode. Often, a hard facing is applied to a tough base material. There is a relatively limited number of studies concerning the machining of objects made of dissimilar materials in the literature. In particular, studies published in [9,10,11,12,13,14,15] are noted. These studies show that processing dissimilar materials entails some unique properties. For example, a monometallic workpiece can be machined from either side, while when machining a workpiece from two different materials, the machining direction must be optimized (for example, machining from the softer material side to the harder material side, or vice versa) [7]. The process of quantifying the surface roughness of an object made of two different metals requires some unconventional parameters (for example, entropy, probability distribution, etc.) [16,17]. The main problem with this uniqueness is the presence of a joint or heat-affected zone, where the composition and properties of the material (especially hardness) differ greatly compared to the constituent materials. The authors in refs. [5,6,7,8,16] described this problem in detail. Depending on whether the cutting tool extends from the softer material side to the harder material side or vice versa, the machining characteristics may differ [17,18]. As a result, the machining forces (cutting force, feed force, etc.) may have a different character when the cutting tool passes the joint zone (Figure 1) [19].

This does not allow rational selection of the cutting tool. The hardness of some bimetals, also in the form of deposited coatings, exceeds the ultimate strength of the cutting tool. Technological difficulties arise in terms of heat removal and thermal conductivity in the cutting zone. Traditional cutting is not possible [17,18,19,20].

It is possible to use copy-piercing electrical discharge machining (EDM) to solve this problem. Currently EDM is one of the most common methods for processing products from modern bimetallic materials. With EDM of products made of bimetallic materials, the physical processes occurring on the surface are to be treated differently from the processes, characteristic of processing products from a homogeneous material [21,22]. This is because the components of the bimetallic material have different physical and mechanical properties and EDM machinability. In [23], the relationship between the processing parameters was established. Bimetal consists of several layers of different materials. This means that the EDM process of complex-shaped products made of bimetals with different machinability will be uneven.

Because the electrical discharge resistance of the elements included in the product made of bimetallic material is different, in the EDM process the electrode-tool (ET) is subject to uneven wear [24,25,26,27]. In the EDM process, not only the processed bimetallic material melts, but also the electrode-tool itself. Uneven machining and uneven tool wear occur. Uneven wear of the tool electrode leads to a decrease in the quality of the machined surface. There is also a decrease in productivity and accuracy [27,28]. Currently, the issue of electrical discharge machining of bimetallic materials has not been fully studied. The use of new materials for electrode-tools for processing bimetals has also not been studied.

Since it is necessary to process a composition of different materials simultaneously in a bimetal using the EDM method, the complexity of the selection of modes arises. It becomes difficult to predict machining stability, accuracy, and quality. EDM workflow development is not applied using technology spreadsheets [25]. Therefore, it is not possible to ensure the quality criteria and processing performance. There is an increase in the time of the technological operation. The cycle of mastering new products increases and production efficiency decreases [28,29,30,31].

An urgent task is to study the EDM process of bimetallic materials. It is important to study the dependence of the influence of EDM modes on the performance of the process and on the formation of quality indicators for the processed surface of products made of bimetallic materials.

The purpose of the work is to increase the efficiency and accuracy of the EDM process of bimetallic products using electrodes-tools with different physical and mechanical properties.

## 2. Materials and Methods

### 2.1. Materials and Methods

A weld coated steel substrate of steel 09G2S and copper M1 (Med Prom Star, Moscow, Russia) as surfacing material was used as a bimetallic workpiece. The bimetallic workpiece was processed in EDM oil of grade IPOL SEO 450 (IPOL Lubricants, Mumbai, India) on an Electronica Smart CNC copy-piercing EDM machine (Electronica Machine Tools, Pune, India). Three different electrodes (ET), namely, graphite, copper, and composite, were used for machining with a constant depth of 5 mm for each run. Table 1 shows the EDM modes used in conducting the experimental work, where: 1. Ip—current strength, A; 2. Ton—pulse action time, μs; 3. U—Voltage, V.

The technique presented in the refs. [32,33] was used to carry out the experimental studies.

The intervals of variation of the factors were chosen based on the technological data specified in the recommendations for electrical discharge machining. The selected intervals of mode variation correspond to the parameters of the equipment of machine-building enterprises using EDM machines. Thus, the dependences obtained in the work are of a practical nature and can be applied in actual engineering production.

The surface morphology of the processed samples was carried out on a scanning electron microscope (SEM) model: Phenom World G2 ProX (Thermo Fisher Scientific, Waltham, MA, USA). According to the methodology presented [34], the magnifications of ×500–4000 and an accelerating voltage of 15 kV were used for the SEM investigation. The surface roughness after EDM was evaluated using a Lext OLS4000 laser scanning microscope (Olympus Corporation, Tokyo, Japan) on a 3D image model obtained using the 3D Roughness Reconstruction software module (Olympus Corporation, Tokyo, Japan). The ET wear was measured on a Carl Zeiss Contura G2 coordinate measuring machine (Carl Zeiss, Oberkochen, Germany).

### 2.2. Modeling

The initial conditions and assumptions of the theoretical model are as follows: the absence of a diffusion layer between dissimilar parts of the workpiece, the ET is homogeneous and makes a translational motion, the properties of the working fluid are the same along the entire length of the gap between the ET and the workpiece.

The processed surfaces of the ET and the workpiece can be specified by implicit functions as:(1)$$F_{e}\left(x,y,z+h,h\right)=0\;F_{w}\left(x,y,z,h\right)=0$$
where *x*, *y*, *z*—Cartesian coordinates of points located on the surface of the ET and the workpiece; $h=\int_{0}^{t}V_{b}(\tau \left(d\tau \right)$—distance traveled by ET along the *z*-axis; $V_{b}\left(\tau \right)$—the speed of ET movement during processing; *t*—time.

Parts of bimetallic products differ in their physical and mechanical properties and in EDM machinability. The material will be removed from each part at varying rates. After a certain time, a certain amount of material is removed from each part. Due to the uneven material removal on each part of the bimetallic workpiece, the value of the interelectrode gap will also differ. In one part of the treatment zone, the value of the interelectrode gap exceeds the value of the breakdown of the working fluid, which stops the processing. In another part of the treatment zone, the process of electrical erosion will continue until the interelectrode gap exceeds the critical value. This critical value corresponds to the breakdown of the working fluid under the given conditions. As a result, a part of the ET surface will wear out more [28]. The EDM machine controls the size of the interelectrode gap, and the machine maintains the size of the gap within a certain range of values. When the discharges are terminated, the ET is supplied by the amount required to resume processing. ET carries out only reciprocating motion. The unevenness of its wear increases during processing. During the EDM process of a bimetallic workpiece, the ET and the workpiece surfaces will acquire a pronounced stepped shape with rounded corners of the profile.

The rounding of the corners of the profile of the ET and the workpiece occurs due to the radial nature of the distribution of the electric field in the interelectrode gap in the nonlinear sections of the processed surfaces. The transitional areas of the treated surfaces acquire a rounded shape. A boundary is formed between the various materials of the bimetallic workpiece at the bottom of the resulting cavity and the corresponding section of the ET. Rounding occurs along the perimeter of the bottom of the resulting blank cavity.

From Equation (1), we derive the equations for changing the surfaces of the workpiece from bimetal and ET during processing [23,24,25,26,27,28].
(2)$$-qKV_{edm}x\left|gradF_{b}\right|+\frac{\partial F_{b}}{\partial h}+\frac{\partial F_{b}}{\partial \left(z+h\right)}=0\;-V_{edm}\left|gradF_{w}\right|+\frac{\partial F_{w}}{\partial h}=0$$
where *q* is the relative volumetric wear of ET; *k* is a coefficient that considers the curvature of the ET); *V_edm_* is the metal removal rate. These equations are shown in Figure 2 for the case of a homogeneous workpiece material.
(3)$$T_{e}\left(x,y,z+h\right)=F_{e}\left(x,y,z+h,h\right)-h\;T_{w}\left(x,y,z\right)=F_{w}\left(x,y,z,h\right)-h$$

Equation (3) are reduced to Eikonal-type equations:(4)$$\left|gradF_{e}\right|=\frac{1}{qkV_{edm}}\left(1+\frac{\partial T_{e}}{\partial \left(z+h\right)}\right)\;\left|gradF_{e}\right|=\frac{1}{V_{edm}}$$

Considering the assumptions made, the theoretical model takes the form of a system of Equation (4) for each part of the workpiece:(5)$$\left|gradF_{e}\right|=\frac{1}{qkV_{edm1}}\left(1+\frac{\partial T_{e1}}{\partial \left(z+h\right)}\right)\;\left|gradF_{w1}\right|=\frac{1}{V_{edm1}}$$
(6)$$\left|gradF_{e2}\right|=\frac{1}{qkV_{edm2}}\left(1+\frac{\partial T_{e2}}{\partial \left(z+h\right)}\right)\;\left|gradF_{w2}\right|=\frac{1}{V_{edm2}}$$
Indices 1 and 2 correlate with sections of the workpiece made of dissimilar materials and the corresponding sections of EI. The EDM speed of the workpiece can be calculated from the relationship:(7)$$V_{edm}=C\rho \lambda T^{2}W_{i}f$$
where: *C* is the heat capacity of the material, *λ* is the coefficient of thermal conductivity, *T* is the melting point, *W_i_* is the pulse energy, *f* is the pulse repetition rate, ρ is the material density.

According to Equation (7), the amount of material removed during one pulse also depends on the pulse energy and on the thermophysical properties of the material being processed. The properties are melting point, thermal conductivity, heat capacity, and density. It is possible to calculate the EDM speed for each of the material components using these properties. When calculating, it must be considered that the EDM is impulsive in nature. The calculation according to the proposed model must be performed according to a cyclic algorithm. This algorithm takes into account the quasi-continuous change in the machined surfaces of the ET and the workpiece.

## 3. Results

Table 2 shows the calculated results for the destruction rate ratio of ET when processing a bimetallic material of the steel–copper type. The destruction rate was the ratio corresponding to the steel part (speed *V_edm1_*) and copper part of the workpiece (speed *V_edm2_*).

The distance traveled by the ET for each section was calculated using the values of the EDM speed of the ET sections from various materials of the workpiece being processed and the constant processing time:(8)$$h=\int_{0}^{t}V_{e}\left(\tau \right)d\tau$$

Table 3 shows the results of the theoretical calculation of material removal when processing steel–copper bimetallic material. Uneven material removal is the difference between the distance traveled by the electrode when processing the steel part of the bimetallic material (*h_stee_*_l_) and the distance traveled by the electrode when processing the copper part (*h_coppe_*_r_).

It was found that with EDM with copper ET, the unevenness of material removal is the smallest in comparison with other ET. To test the theoretical model, experiments were carried out to study the wear of the ET and the unevenness of the material removal during the EDM of a bimetallic material of the steel–copper type by dissimilar ET.

Figure 3 shows the uneven material removal during processing with three types of ET: graphite, copper, and composite ET in the medium mode, where H is the amount of uneven material removal.

Summary data for all processing modes for dissimilar electrodes are shown in Table 4.

With EDM ET from graphite of a bimetallic material of the steel–copper type, the greatest uneven material removal is achieved at med and max modes and is 3.2–3.5 mm. Graphite ET wears out intensely during EDM. When processing a bimetallic sample with copper EI, the maximum uneven material removal is achieved at the maximum mode and is 2.9 mm. The minimum uneven material removal is 1.4 mm, achieved at min modes. It is necessary to select the minimum processing modes when processing materials with high thermal conductivity. Heat is distributed evenly throughout the workpiece. The part does not overheat.

When processing with composite ET, the greatest uneven material removal was achieved at the maximum EDM mode, which is 1.9 mm. It is advisable to process bimetallic material of the steel–copper type at the med and max modes, obtaining the least uneven processing at a higher productivity. The efficiency of processing with composite ET is related to the structure and physical and mechanical properties of the composite ET. An increase in the amount of additives in a composite ET with high electrical resistance (chromium, tungsten, molybdenum) and at the same time an increase in porosity lead to an increase in the electrical resistivity of the composite material in proportion to the amount of refractory metal additive. The features of the formation of the structure of composite ET are considered in more detail in [34,35,36].

The analysis of the obtained results showed that the machinability of copper and steel depends on the processing modes and properties of the ET. As the pulse energy increases, the performance increases. However, when processing a bimetal, there is an uneven processing of each of the components of the bi-metal. The appearance of a step is noted. This suggests that in the process of processing the copper component of the bimetal, the intensity of wear of the ET occurs.

From a physical point of view, the occurrence of unevenness during processing occurs due to the uneven transfer of energy from the spark discharge to the electrodes. Because the thermophysical properties of the components of the bimetal are different, there is an uneven distribution of the electric field in the breakdown channel and, as a consequence, the uneven formation of thermal energy. According to scientific data [35,36], the energy supplied to the electrodes is total and consists of electronic, ionic, torch, gas-kinetic, radiant, and volumetric components. Flame, electronic, and ionic components have the greatest significance for material removal. It is important to note that the torch transfer of the processed material occurs in the vapor state in the presence of a temperature difference between the electrodes. Thus, when processing materials with different thermophysical properties, uneven formation of the total energy components occurs. It is worth noting the influence of the thermophysical properties of materials on their electroerosive resistance. An important factor characterizing the electrical erosion resistance of a material is its electrical conductivity. Thus, the copper component of the bimetallic material is characterized by increased electroerosive resistance. In this connection, the result of the experiment becomes adequate, showing in all experiments the greatest wear of the ET during processing of the copper component of the bimetal.

In Figure 3a, when using graphite EI, the greatest processing unevenness is observed. This phenomenon can be explained by the influence of the “torch” component of the flare discharge. In the material of the ET with a high content of graphite, a larger amount of the vapor phase is formed, as a result of which there is an uneven processing of the bimetal components.

The use of a composite electrode allows for the greatest uniformity of processing. Erosion of a graphite-based composite electrode leads to decomposition (pyrolysis) of the electrode surface layer, forming charged carbon particles in the gap, these particles are attracted back to the electrode, creating a passive layer of pyrolysis graphite. In this way, it is possible to maintain a sufficiently high productivity and eliminate or minimize the wear of the ET. Comparison of the values of uneven removal of steel–copper bimetallic material obtained by the theoretical method and the values obtained by the practical method showed a discrepancy of about 15%.

In addition to the bimetallic material of the steel–copper type, the formation of unevenness also occurs on the ET. Figure 4 shows a diagram of the formation of unevenness on the graphite ET during processing in the med-mode, where h is the unevenness of the ET wear. The summary data of the wear of dissimilar ET for all processing modes are shown in Table 5.

Wear of ET is observed in EDM with graphite ET in the med and max modes. Increased wear of graphite ET and low processing efficiency are caused by its low mechanical strength.

EDM with copper ET showed relatively equal results of ET wear in all modes. In addition to uneven material removal, one can see growths on the ET. These build-ups significantly impair the quality of EDM. It primarily depends on the thermophysical properties of the material—the melting temperature and thermal conductivity. Composite EDM allows high productivity of EDM to be ensured, as well as a stable process flow. With EDM with copper ET at the modes med and max, ET wear is minimal and amounts to 1–1.1 mm, respectively.

In this work, a study of the roughness parameter of the processed surface of a steel–copper bimetallic material after processing with dissimilar ET in three modes was carried out. The results of measuring the roughness are presented in Table 6.

Figure 5 shows the relief of the processed surface of the steel material in the med mode of processing with three dissimilar ET.

A distinctive feature of the processed steel surface of the bimetallic material is the microcracks along the grain boundaries. When processing a steel layer, the largest number of microcracks is observed during processing with steel ET. Processing with copper and composite ET leads to the formation of chip sand build-ups, as well as an increase in the size of microcracks on the treated surface.

After the surface treatment of the steel of the bimetallic material with composite ET, traces of the solidified ET material are visible. Such deposited particles of copper ET cover the steel surface unevenly. One can notice the formation of microcracks on the solidified material of the electrode-tool.

The presence of such microcracks on the surface of the steel layer can lead to the occurrence of macro-defects in this layer and a decrease in the performance properties of a part made of this material.

Figure 6 shows the relief of the processed surface of the copper material in the med mode of processing with three dissimilar ET.

A distinctive feature of the processed surface of copper of a bimetallic material is the presence of micro-roughnesses that form a general structure of the fracture surface in the form of a cobweb with combined large holes with a cubic structure. The formation of such elements on the copper surface of a bimetal after EDM is complex.

It is shown that the processing of a copper bimetallic layer with graphite ET leads to the formation of a structure of the treated surface in the form of a spider’s web and the formation of united small-sized holes with a cubic structure. Processing in this mode of copper ET leads to an increase in the size of the combined wells. The largest dimensions of such holes on the treated copper surface can be observed after processing with composite ET.

## 4. Conclusions

1. A theoretical model was developed that allows the calculation of the amount of removal of bimetallic material of steel–copper depending on the EDM modes and the ET material. This model allows one to calculate the EDM speed for each of the material components using these properties. The model takes into account the impulsive nature of the EDM process, as well as the quasi-continuous change in the processed surfaces of the ET and the workpiece. The convergence of the theoretical model with the results of experimental studies is 15%.

2. An experimental study of the ET wear during EDM of a steel–copper bimetallic material was carried out depending on the EDM modes and the ET material. It was found that with EDM with composite ET at the mode min, ET wear is minimal and amounts to 1mm.The use of composite ET allows for the greatest uniformity of processing. Erosion of the graphite-based composite ET leads to decomposition (pyrolysis) of the electrode surface layer, forming charged carbon particles in the gap, these particles are attracted back to the electrode, creating a passive layer of pyrolysis graphite. In this way, it is possible to maintain a sufficiently high productivity and minimize the wear of the ET.

3. The non-uniformity of the properties of the material ET and the workpiece, as well as the increased erosion resistance of ET relative to the workpiece, leads to the uneven formation of the components of the total energy. A high concentration of flare energy in the areas of processing leads to the destruction of the surface and the appearance of cracks. The calculation of the quality parameter and analysis of the quality of the processed surface of the steel–copper EDM bimetallic material in different modes of processing ET with different electrophysical properties, found that during the processing of the steel layer, regardless of the ET material, microcracks are formed along the grain boundaries, while during the processing of the copper layer, the formation of enlarged holes occurs. These phenomena lead to a deterioration in the operational properties of the workpiece.

## Figures and Tables

**Figure 1 materials-15-00750-f001:**
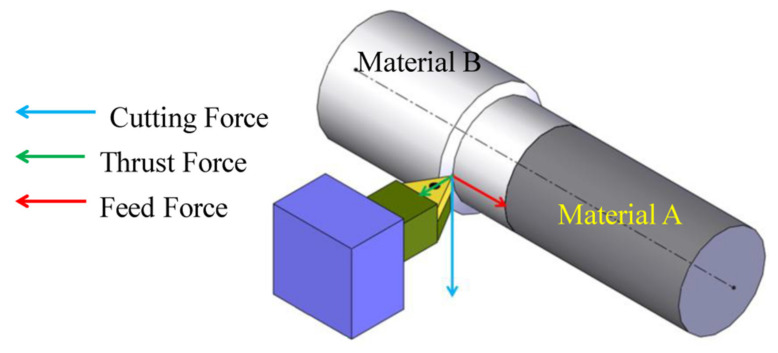
Feature of electrical discharge machining of bimetallic material with a blade tool.

**Figure 2 materials-15-00750-f002:**
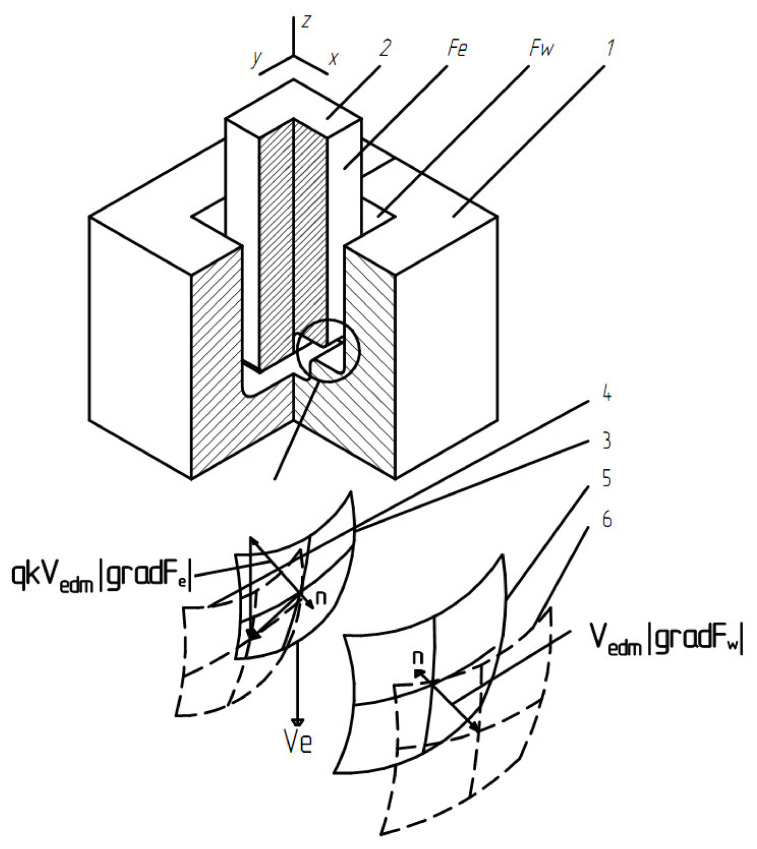
Changing the surfaces: 1—blank; 2—electrode tool; 3—the initial surface of the electrode; 4—a modified surface of the electrode; 5—the initial surface of the workpiece; 6—changed surface of the workpiece.

**Figure 3 materials-15-00750-f003:**
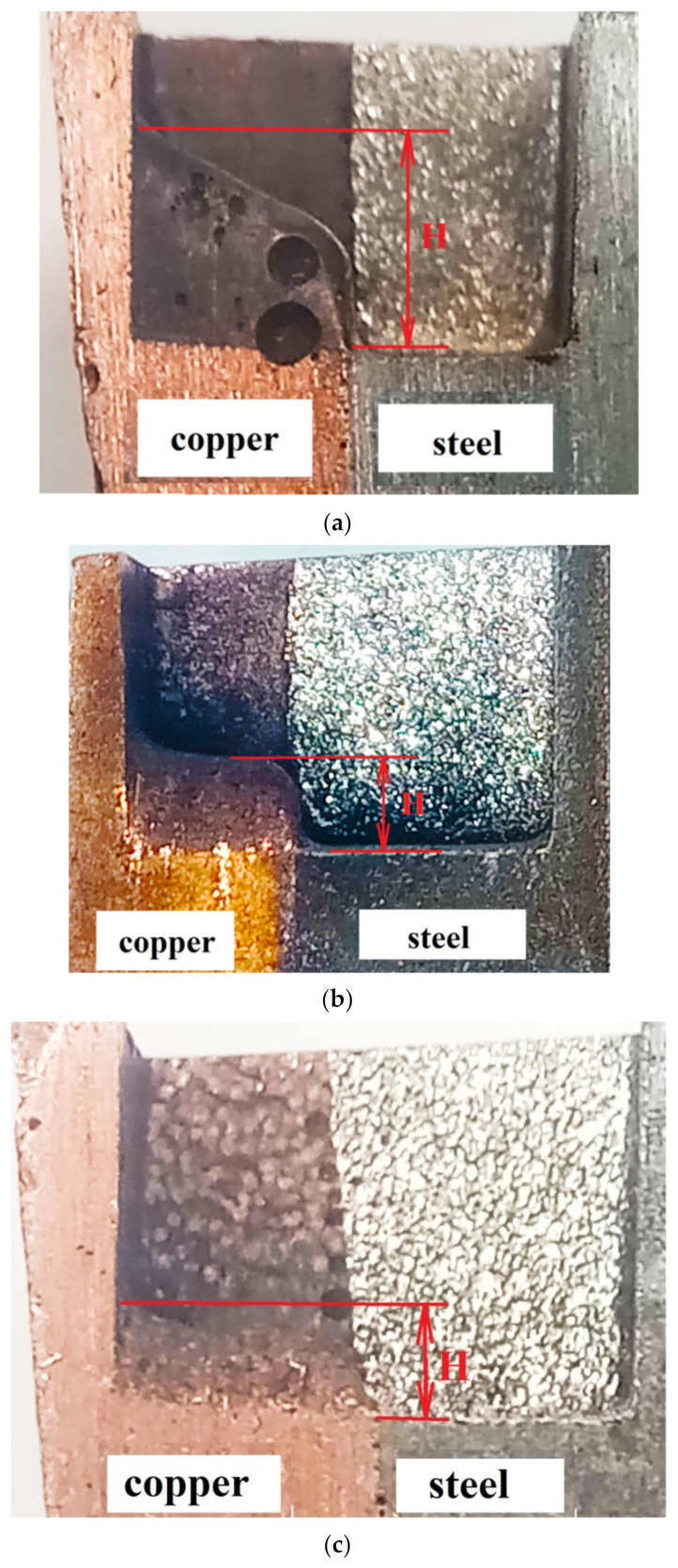
The unevenness of material removal during processing with three types of EI in the middle mode: (**a**) graphite ET, (**b**) copper ET, (**c**) composite ETI.

**Figure 4 materials-15-00750-f004:**
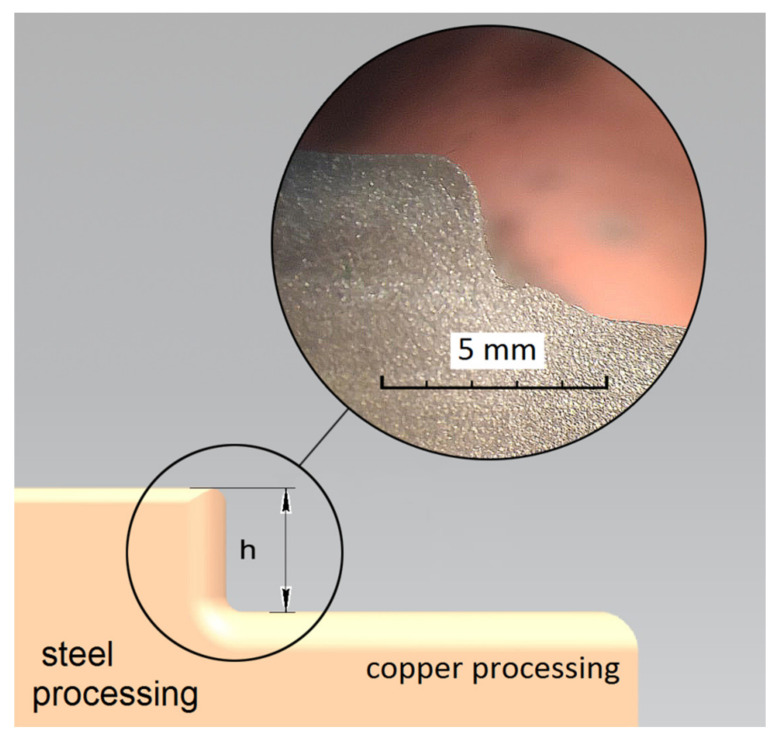
Uneven wear of ET during graphite EI processing in medium mode.

**Figure 5 materials-15-00750-f005:**
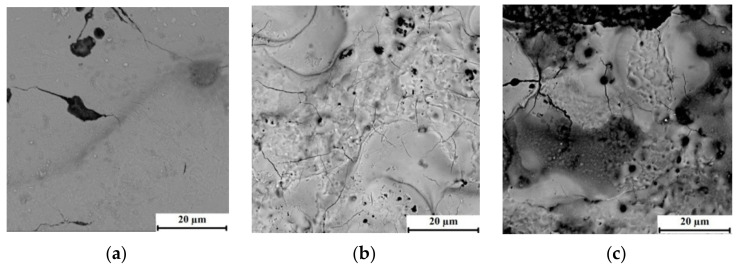
The processed surface of the steel layer of bimetallic material (×4000): (**a**) graphite ET, (**b**) copper ET, (**c**) composite ET.

**Figure 6 materials-15-00750-f006:**
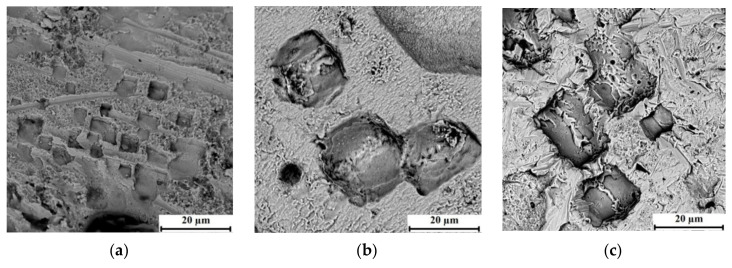
The processed surface of the copper layer of bimetallic material (×4000): (**a**) graphite ET, (**b**) copper ET, (**c**) composite ET.

**Table 1 materials-15-00750-t001:** Processing modes for bimetallic material [32,33].

Mode	Ton (μs)	Ip (A)	U (V)
Min	1	0.5	50
Med	100	3	50
Max	750	20	50

**Table 2 materials-15-00750-t002:** The ratio of the destruction rate of the electrode sections.

Processing Mode	Graphite ET	Copper ET	Composite ET
	*V_edm1/_V_edm2_*	*V_edm1/_V_edm2_*	*V_edm1/_V_edm2_*
Min	0.10	0.04	0.07
Med	0.23	0.18	0.03
Max	0.24	0.15	0.04

**Table 3 materials-15-00750-t003:** The calculated values of the unevenness of material removal during processing of steel–copper bimetallic material, mm.

Processing Mode	Graphite EI	Copper EI	Composite EI
	*h_steel_-h_copper_* _,_ _mm_	*h_steel_-h_copper_* _,_ _mm_	*h_steel_-h_copper_* _,_ _mm_
Min	1.9	3.1	3.3
Med	1.7	1.8	2.7
Max	1.3	1.4	1.7

**Table 4 materials-15-00750-t004:** Uneven material removal during processing with heterogeneous EI in three modes, H, mm.

ET Material	H, Min Mode	H, Med Mode	H, Max Mode
graphite	2	3.2	3.5
copper	1.8	1.9	2.9
composite	1.4	1.5	1.8

**Table 5 materials-15-00750-t005:** Uneven material removal during processing with heterogeneous ET in three modes, H, mm.

ET Material	h, Min Mode	h, Med Mode	h, Max Mode
Graphite	1.5	2.8	3
Copper	1.2	1.5	2
Composite	1	1.1	1.3

**Table 6 materials-15-00750-t006:** Roughness parameter of bimetallic material when processing heterogeneous ET in three modes.

Material ET	Min Mode		Med Mode		Max Mode	
	Ra on Copper (μm)	Ra on Steel (μm)	Ra on Copper (μm)	Ra on Steel (μm)	Ra on Copper (μm)	Ra on Steel (μm)
Graphite	1.8	0.8	2.7	1.3	3.6	1.9
Copper	4	1.9	6.1	3.2	8.9	4.4
Composite	7	3	9.1	4.6	12.2	6.1

## Data Availability

Not applicable.

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
