# Peer review of "Study of the EDM Process of Bimetallic Materials Using a Composite Electrode Tool"

_materials, 2022, doi:10.3390/ma15030750_

Round 1

Reviewer 1 Report

In this paper, authors have tried to machine bimetals by electrical discharge machining (EDM) with graphite, copper, and composite electrodes. They studied the dependence of the influence of EDM modes on the performance of the process, on the formation of quality indicators for the processed surface of products made of bimetallic materials. They aimed to increase the efficiency and accuracy of the EDM process of bimetallic products using electrode-tools with different physical and mechanical properties. The study has significance because EDM of bimetallic materials is still not thoroughly explored.

However in the discussion part, it is found that rather than explaining the causes of the particular trend, the authors are merely highlighting the facts they observed during experimentation. They need to explain why a particular effect is happening. The experimentation part is OK, but the conclusions are again merely reiterating the experimentations. So the paper needs corrections in light of these points.

  1. Why did you select the range of process parameters as given in Table 1? Please explain
  2. The reasons for unevenness of material removal during processing needs more explanation.
  3. In most of the explanations, only quantification of data already shown in tables is found. Please also include the reasons for obtaining that particular trend of data.
  4. The conclusions need rewriting. They should be crisp and precise.

Author Response

Dear Reviewer,

I am grateful for the helpful and interesting comments by you. The comments have been addressed in the following way (Changes are highlighted in RED color in the manuscript.).

Reviewer 2 Report

Comments on the paper proposed by Ablyaz et al: “Study of the EDM process of bimetallic materials using composite electrode tool”.

Here are reported some considerations:

  • the abstract is too long reporting information that must be reported in the paragraph of the introduction. In general, the abstract briefly reports the reasons for the study, the innovative aspect and the experimental methods used.

  • The introduction should be rewritten in a more abbreviated way. In addition, there are several errors and non-meaningful sentences. In particular, from lines 47 to 51 excessive punctuation was used, using too many periods to formulate a concept.

  • It would be better to remove figure 1 from the introduction and insert it in the following paragraphs. In general, figures are not included in the introduction unless strictly necessary.

  • In order to shorten the introduction, the sentence on lines 81-84 could be deleted because it does not add essential information.

  • The innovative nature of the work with respect to scientific literature should also be emphasized in the introduction.

6) Move the quotations [2-5] to line 46 (it makes no sense where they were inserted). In place of [2-5] insert the following citations at line 44:

Shakerin, Sajad, et al. "Additive manufacturing of maraging steel-H13 bimetals using laser powder bed fusion technique." Additive Manufacturing 29 (2019): 100797.

Epasto, G., et al. "Experimental investigation of rhombic dodecahedron micro-lattice structures manufactured by Electron Beam Melting." Materials Today: Proceedings 7 (2019): 578-585.

Liu, Liming, et al. "Additive manufacturing of steel – bronze bimetal by shaped metal deposition: interface characteristics and tensile properties." The International Journal of Advanced Manufacturing Technology 69.9-12 (2013): 2131-2137.

7) Line 40 talks about new materials, but what do we mean with new materials? Explain better

8) Paragraph 2.1 talks about both materials and methods. It needs to be renamed.

9) What do the parameters in table 1 represent? They should be described

10) The numbering of the formulas is wrong. On lines 180-183 there are 4 different formulas that must be numbered.

11) On lines 188-189 there are typos. (Add subscript and insert density).

12) Rewrite lines 194-196. Why is this punctuation used?

13) Line 199 there are some typos

14) In table 2-3 the units are missing. Also is used both a period and a comma.

15) Figure 3 (c) has a very bad quality; can it be replaced?

16) On line 262 what is meant by "significant destruction"?

17) Line 267 what does "T" represent?

18) In the conclusions the authors argue that between the experimental method and the analytical method there is a difference of 15%. From comparisons with the scientific literature, can this be called an accurate method? Explain better

For all the previous reasons, the reviewer recommends major amendments of paper for publication in Materials.

Author Response

Dear Reviewer,

I am grateful for the helpful and interesting comments by you. The comments have been addressed in the following way (Changes are highlighted in RED color in the manuscript.)

Round 2

Reviewer 2 Report

The article has been improved by the authors following the suggestions of the reviewers. For this reason publication is recommended.